# BetterBodies: Reinforcement Learning guided Diffusion for Antibody Sequence Design

## Abstract

Antibodies offer great potential for the treatment of various diseases. However, the discovery of therapeutic antibodies through traditional wet lab methods is expensive and time-consuming. The use of generative models in designing antibodies therefore holds great promise, as it can reduce the time and resources required. Recently, the class of diffusion models has gained considerable traction for their ability to synthesize diverse and high-quality samples. In their basic form, however, they lack mechanisms to optimize for specific properties, such as binding affinity to an antigen. In contrast, the class of offline Reinforcement Learning (RL) methods has demonstrated strong performance in navigating large search spaces, including scenarios where frequent real-world interaction, such as interaction with a wet lab, is impractical. Our novel method, BetterBodies, which combines Variational Autoencoders (VAEs) with offline RL guided latent diffusion, can generate novel sets of antibody CDRH3 sequences from different data distributions. Furthermore, we reflect biophysical properties in the VAE latent space using a contrastive loss and add a novel Q-function based filtering to enhance the affinity of generated sequences. Using the Absolut! simulator, we demonstrate that BetterBodies generates sequences with improved binding affinity to the SARS-CoV spike receptor-binding domain and matches or outperforms the state-of-the-art method Generative Flow Network (GFlowNet). In conclusion, our method has the potential for great implications in real-world biological sequence design, where the generation of novel high-affinity binders is a cost-intensive endeavor.

## 1 Introduction

Antibodies are a class of proteins with great potential for treating diseases such as cancer (Kaplon et al., 2023; Norman et al., 2020; Robert et al., 2022). However, the discovery of therapeutic antibodies in classical wet lab experiments is constrained by high costs and low throughput (Angermueller et al., 2019; Angermüller et al., 2020; Shanehsazzadeh et al., 2023). Computational antibody design using generative models, therefore, holds immense potential for reducing the time and resources needed (Shanehsazzadeh et al., 2023).

Diffusion models have recently received considerable attention due to their ability to generate diverse and high-quality data (Murphy, 2023). Their versatility makes them applicable to numerous tasks in the realm of protein design, including protein structure prediction (Anand & Achim, 2022), protein-protein docking (Ketata et al., 2023), and protein design (Chen et al., 2024; Watson et al., 2023). However, basic diffusion is not capable of optimizing for a desired property such as binding affinity to an antigen. In contrast, RL methods have demonstrated remarkable efficacy in identifying solutions in large search spaces (Silver et al., 2016). In the domain of offline RL, the objective is to learn an optimal policy from a pre-collected dataset without any real-world interaction. This is well-suited to antibody sequence design, where direct access to a wet lab is not feasible. Consequently, the combination of diffusion models and (offline) RL methods has great potential for the field of computational antibody design. Wang et al. (2023) demonstrated that RL methods can be utilized to guide continuous diffusion models toward optimal regions within the explored space.

In our work, we integrate several of these recent advances into an approach we term *BetterBodies* to design novel diverse antibody CDRH3 sequences with improved binding affinity given a set of training sequences. We introduce a VAE to map amino acids (AAs) into a continuous latent representation, allowing us to apply the work by Wang et al. (2023) to the AA sequence design task. The offline RL policy, represented by a continuous diffusion model guided by a learned Q-function, is then trained to stepwise diffuse AAs in this latent space to generate novel sequences. This stepwise approach facilitates the stitching of parts of suboptimal sequences contained in the training data to create improved sequences (Kumar et al., 2022). Furthermore, we demonstrate how to improve the affinity of generated sequences by exploiting properties of both the learned AA latent space and the learned Q-function. Therefore, we introduce a contrastive loss (Khosla et al., 2020) to shape the VAE latent space to reflect biophysical properties and show that the learned Q-function can be used as a discriminator to sort out low-affinity sequences. In experiments on the Absolut! benchmark (Robert et al., 2022), we demonstrate that BetterBodies can learn from a variety of distributions of sequences and affinity values, including random sequences, sequences generated by an RL agent, and murine antibody sequences. Thereby, sequences generated using BetterBodies match or outperform those generated by the state-of-the-art algorithm GFlowNet Jain et al. (2022).

## 2 BACKGROUND

In this section, we provide the necessary background on antibody sequence design, VAEs, latent diffusion models, and RL.

### 2.1 ANTIBODY SEQUENCE DESIGN

Antibodies are a class of proteins, consisting of a sequence of AAs, utilized by the immune system to recognize and bind foreign molecules (antigens) with high specificity (Norman et al., 2020; Robert et al., 2022). Due to their favorable binding properties, they have become the leading class of new drugs developed (Lu et al., 2020; Norman et al., 2020).

Given that there are 20 possible AAs to be placed at each sequence position, the total search space for sequences of length $L$ contains $20^L$ sequences. However, it has been demonstrated that specific regions of the antibodies, the so-called Complementarity Determining Regions (CDRs), contain the majority of antigen-binding AAs (Norman et al., 2020). Furthermore, the third CDR of the heavy chain (CDRH3) has been shown to have the largest influence on the antibodies' specificity (Xu & Davis, 2000). Consequently, we utilize the design of CDRH3 sequences as a proxy for the design of complete antibodies. The Absolut! software, which we employ to approximate antibody CDRH3 binding affinity to an antigen, fixes the length of this region to 11 positions. Thus, in this work, we will set the length of designed sequences to $L = 11$, resulting in approximately 205 trillion possible sequences. This vast space precludes exhaustive search, thereby underscoring the potential impact of computational antibody design on wet labs.

### 2.2 VARIATIONAL AUTOENCODERS

Autoencoders are encoder-decoder networks trained to minimize a reconstruction loss between their input $x$ and reconstructed input $d_\rho(e_\omega(x))$, where $e_\omega$ is the encoder network and $d_\rho$ is the decoder network represented by their learnable parameters $\omega$ and $\rho$. The VAE (Kingma & Welling, 2014) is a specific type of autoencoder in which the continuous latent representation, denoted by $z = e_\omega(x)$, follows a probabilistic distribution $p_\omega(z|x)$. The latent representation $z \sim \mathcal{N}(\mu_\omega^x, \sigma_\omega^x)$ is typically defined as a Gaussian distribution with a learned mean $\mu_\omega$ and standard deviation $\sigma_\omega$. In addition to the reconstruction loss, the VAE is regularized such that the latent distribution minimizes the Kullback-Leibler (KL) divergence to a Gaussian distribution $\mathcal{N}(0, \mathbf{I})$, facilitating a dense latent space. In our setting, we use VAEs to encode AAs classes into a two-dimension latent space and use a Binary Cross Entropy loss for reconstruction. Furthermore, VAEs can be regularized to cluster inputs belonging to the same class by pulling them together in embedding space, while simultaneously pushing apart clusters of inputs from different classes (Khosla et al., 2020).

## 2.3 DIFFUSION MODELS

Diffusion models employ a *forward process*, or *diffusion process*, to gradually corrupt observed data into noisy data and learn a *reverse process*, or *denoising process*, to undo the corruption. A trained model can thus be used to generate high-quality data from noise (Murphy, 2023).

In this work, we are dealing with both diffusion steps $n \in \{0, .., N\}$ and time steps $t \in \{0, ..., T\}$. To facilitate clarity, we will use superscripts for diffusion steps and subscripts for time steps. Diffusion probabilistic models (Ho et al., 2020; Sohl-Dickstein et al., 2015) are a class of latent variable models defined as $p_\theta(x^0) := \int p_\theta(x^{0:N}) dx^{1:N}$. Here, $x^1, ..., x^N$ are latent variables of the same dimensionality as the data sample $x^0$ drawn from the observed data distribution $q(x^0)$. In our setting, these data samples are two-dimensional embeddings of AAs drawn from a VAE latent space. The forward process gradually adds Gaussian noise to $x^0$ according to a noise schedule $\beta^1, ..., \beta^N$, over $N$ steps (Ho et al., 2020). In particular, the forward process is defined as $q(x^{1:N}|x^0) := \prod_{n=1}^{N} q(x^n|x^{n-1})$, with single step transition $q(x^n|x^{n-1}) := \mathcal{N}(x^n; \sqrt{1-\beta^n}x^{n-1}, \beta^n \mathbf{I})$.

The reverse process is the joint distribution $p_\theta(x^{0:N})$ defined as a Markov chain starting at $p(x^N) = \mathcal{N}(x^N; 0, \mathbf{I})$ given as $p_\theta(x^{0:N}) := p(x^N) \prod_{n=1}^{N} p_\theta(x^{n-1}|x_n)$, with a learned Gaussian transition $p_\theta(x^{n-1}|x^n)$. The objective of training $p_\theta$ is to maximize the expected log-likelihood of the data, given by the evidence lower bound (ELBO) $\mathbb{E}_q[\log \frac{p_\theta(x^{0:N})}{q(x^{1:N}|x^0)}]$. In essence, the objective is to maximize the probability of reconstructing a sample $x^0$ from a noisy sample $x^N$. In practice, instead of predicting $x^{n-1}$ given $x^n$, a noise prediction model $\epsilon_\theta$ is trained (Ho et al., 2020; Murphy, 2023). Consequently, the loss for the diffusion model given a dataset $D$ can be simplified to

$$L(\theta) = \mathbb{E}_{n \sim \text{Unif}(1,N), \epsilon \sim \mathcal{N}(0,\mathbf{I}), x^0 \sim D}[||\epsilon - \epsilon_\theta(\sqrt{\bar{\alpha}^n}x^0 + \sqrt{1-\bar{\alpha}^n}\epsilon, n)||^2], \quad (1)$$

where $\alpha^n := 1 - \beta^n$ and $\bar{\alpha}^n := \prod_{i=1}^{n} \alpha^i$.

## 2.4 REINFORCEMENT LEARNING

In RL, tasks are typically formulated as Markov Decision Processs (MDPs). We define a deterministic MDP as a tuple $\langle S, S_0, A, P, R \rangle$, where $S$ is the set of possible states, $S_0$ is the set of initial states $S_0 \in S$, $A$ is the set of possible actions $a$ executable in $s \in S$, $P$ is a deterministic transition function $P(s, a) : S \times A \mapsto S$, and $R$ is a deterministic reward function $R(s, a) : S \times A \mapsto \mathbb{R}$.

In this work, we address the task of designing discrete AA sequences, representing antibody CDRH3 sequences targeting a specific antigen. We choose to frame the task as a stepwise generation process where the AAs are placed in the sequence one after the other. To evaluate the binding affinity of designed sequences given a specific antigen, we utilize Absolut! (Robert et al., 2022) which sets the length of a *complete* sequence to 11. Thus, we define the set of states $S$ as the set of all possible AA sequences up to length 11, including the empty sequence. We define the set $S_0$ as an empty sequence. The set $A$ is then defined as the set of 20 natural AAs. To facilitate notation, the symbol $a$ is used to refer to the action, the AA it represents, and its two-dimensional VAE latent representation. Consequently, we define $P(s, a) = s \parallel a$ as the concatenation of the sequence generated thus far with the next AA, extending the sequence by one more AA. The reward function $R(s, a)$ is defined corresponding to the predicted free energy using the Absolut! software. As sequences of length shorter than 11 can not be evaluated, the reward function is sparse, returning the evaluated free energy Absolut! $(s\|a|\text{antigen})$ for sequences of length 11 and a reward of 0 for all shorter sequences.

The objective in RL is to learn a policy $\pi$ that maximizes the expected sum of rewards. The action-value function $Q$ represents this expected sum starting from a given state $s_t$. We define it as follows:

$$Q(s_t, a_t) = \mathbb{E}_\pi[R(s_t, a_t) + \sum_{i=1}^{10-t} R(s_{t+i}, a_{t+i})|a_{t+i} \sim \pi(s_{t+i})]. \quad (2)$$

The policy $\pi$ should thus select the action $a$ that maximizes $Q$ for each state $s$. As the search space of CDRH3-sequences is enormously huge, we estimate $\pi$ and $Q$ with function approximations $\pi_\theta$ and $Q_\phi(s, a)$, parameterized by $\theta$ and $\phi$ respectively.

In this work, we focus on the offline RL setting, where the agent is trained using a pre-collected dataset, which we consider to be more realistic for the antibody design task, as interactive access to

a wet lab is not feasible. The offline setting presents its own set of challenges (Levine et al., 2020), mainly erroneous assignment of high Q-values to state-action pairs outside the provided dataset and a resulting distribution shift in the policy. There are multiple approaches to prevent these issues. Our method falls in the class of policy regularization, providing an incentive to remain close to the provided dataset.

## 3 RELATED WORK

In recent years the field of protein sequence design has been approached with a variety of generative models and from a multitude of perspectives. Some (Cowen-Rivers et al., 2022; Khan et al., 2022; Vogt et al., 2023) approach the task in an online setting, where the policy has continuous access to the evaluation metric and can freely explore the design space to find a high-reward sequence using RL or bayesian optimization methods. In other settings, which are sometimes referred to as active learning settings, the generative policy is trained from pre-collected offline datasets for multiple rounds where generated sequences can be evaluated and might be added to the datasets in between rounds (Angermueller et al., 2019; Angermüller et al., 2020; Jain et al., 2022). In such settings, ensembles of evolutionary algorithms (Angermüller et al., 2020), RL algorithms (Angermueller et al., 2019), and GFlowNets (Jain et al., 2022) have been employed as generative models. Lastly, the task can also be approached from a purely offline perspective, where the generative policy is trained only once on a pre-collected offline dataset and then evaluated (Chen et al., 2024; Gruver et al., 2024; Jain et al., 2022).

We approach the task from a purely offline perspective and will present related work from that domain in more detail here. In some of their experiments, Jain et al. (2022) utilize GFlowNets to tackle the design of DNA sequences and protein sequences in the offline setting. They thereby utilize a learned reward model to explore beyond the offline data. In addition to generating samples that optimize a desired property, the networks are also trained to generate samples with high uncertainty according to the learned reward model. The choice of GFlowNets, which are trained to generate samples with likelihoods proportional to their reward fraction in the dataset, intuitively allows for the generation of high-reward samples. In practice, this class of networks is hard to train, due to oversampling of low reward trajectories, and the rewards need to be non-linearly scaled to achieve a good performance (Jain et al., 2022; Shen et al., 2023).

In their approach, Chen et al. (2024) utilize a continuous diffusion model to generate entire antimicrobial peptide (AMP) sequences in an ESM-2 (Lin et al., 2023) latent space. The choice of diffusion models, which are capable of modeling complex multi-modal distributions (Ho et al., 2020; Wang et al., 2023), appears well suited for the complex circumstances underlying AA sequence design. They demonstrated that generated peptides exhibited similar physicochemical properties to natural peptides and aligned closely with respect to AA diversity, which highlights the expressive power of their method. However, they do not employ any technique guiding the diffusion process towards improved sequences. In contrast, Gruver et al. (2024) employ discrete diffusion, whereby sequences are directly diffused in the discrete sequence space. They propose guiding the diffusion model by a learned value function. However, their formulation requires the diffusion model and the value function to share some of their hidden layers and requires the value function to be trained on corrupted inputs (Gruver et al., 2024). Furthermore, wet lab experiments were conducted on generated antibody sequences, which indicated that some of the designed sequences may have improved real-world binding affinity.

RL methods have been demonstrated to identify solutions in large search spaces (Silver et al., 2016) and to be applicable to the sequence design task (Angermueller et al., 2019; Cowen-Rivers et al., 2022; Vogt et al., 2023). Moreover, it has been shown that learned Q-functions, employed in RL, can be utilized to guide continuous diffusion models towards high rewards in offline RL (Wang et al., 2023).

In our work, we apply recent advances in offline RL to the protein sequence design task. Similar to Jain et al. (2022) but in contrast to Gruver et al. (2024) and Chen et al. (2024) we choose to phrase the protein sequence as a stepwise AA generation task, conditioned on the sequence generated thus far, instead of generating entire sequences at once. Such an approach facilitates the use of Q-functions which are able to stitch together improved sequences from suboptimal ones (Kumar et al., 2022). We enable continuous diffusion of discrete AAs by training a VAE encoding discrete AA into

a latent space and decoding generated continuous vectors back to the discrete AAs. Furthermore, we show how biophysical properties can be injected into the latent space to improve performance. Finally, we propose a novel filtering method based on learned Q-values to enhance the average affinity in the set of returned sequences.

# 4 BetterBodies

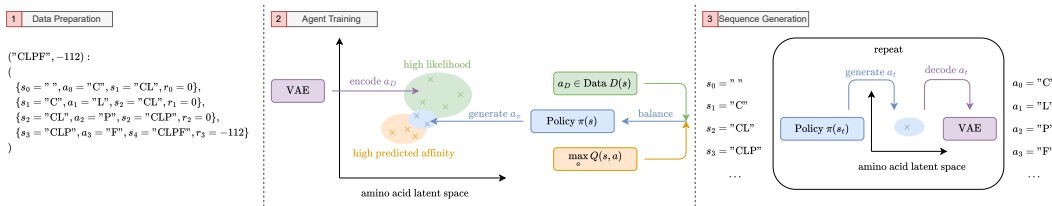

Figure 1: Overview over our method on a fictional sequence of length 4. (1) A given dataset comprising sequence-affinity pairs is transformed into subsequences ($s$) and actions ($a$) which extend those sequences with additional amino acids, together with rewards representing the affinity of the full sequences. (2) Our method utilizes a VAE to encode AAs into a two-dimensional latent space. The diffusion policy $\pi$ is trained to generate a latent AA vector $a_\pi$ given an incomplete amino acid sequence $s$. We balance the policy between generating AAs with high likelihood given the training dataset $D$ and AAs that maximize a learned Q-function, which predicts sequence affinity to a given antigen. (3) By repeating the generative process, AAs are iteratively concatenated to generate a sequence. In each timestep $t$ the policy $\pi$ generates a latent vector $a_t$ given $s_t$. Subsequently, the VAE decodes the AA, which is then concatenated to $s_t$ to generate $s_{t+1}$.

The objective of our method, BetterBodies, summarized in Figure 1, is to train a policy $\pi$ that, in a stepwise manner, generates high-affinity CDRH3 sequences by concatenating generated AAs given an initial set $D$ of sequence-affinity pairs. Furthermore, the generated sequences should be novel and diverse. Diffusion models have recently gained popularity due to their ability to model complex distributions and generate diverse and high-quality samples (Ho et al., 2020; Murphy, 2023; Wang et al., 2023). Consequently, we represent the policy $\pi_\theta$ using a continuous latent diffusion model with parameters $\theta$.

## 4.1 Continuous Amino Acid Representations and Encoding Biophysical Properties

In contrast to discrete diffusion (Gruver et al., 2024), which is designed to directly generate categorical values in the reverse process, our policy $\pi$ generates a continuous latent vector. This continuous vector facilitates guiding the diffusion model and shaping the latent space to incorporate biophysical properties but requires representing the categorical AAs as continuous vectors. We choose to represent them using the two-dimensional latent vectors of a VAE which we train and freeze before training the diffusion model. The latent vectors generated by $\pi$ are then transformed back to discrete AAs using the VAE's decoder network. To train the VAE, each AA $a$ is represented as a one-hot vector and mapped to a two-dimensional latent $z = e_\omega(a) \sim \mathcal{N}(\mu_\omega^a, \sigma_\omega^a)$ using the encoder network $e_\omega$. The decoder network $d_\rho(z)$ then maps the latent vector $z$ back to a probability distribution over discrete AAs. Consequently, the VAE can be trained end-to-end by minimizing the Binary Cross Entropy (BCE) loss between the input $a$ and the decoder's output. Additionally, the distribution of latent variables $z$ is regularized to minimize the KL divergence to the Gaussian distribution. The loss function of the VAE is then given as $L(a) = \text{BCE}(a, d_\rho(e_\omega(a))) + \text{KL}(\mathcal{N}(\mu_\omega^a, \sigma_\omega^a), \mathcal{N}(0, \mathbf{I}))$.

Furthermore, the latent space utilized in our method also allows for incorporating additional biases. As a proof of concept, we chose to regularize the VAEs latent space to represent AA properties. Specifically, we group the AAs according to their side chains' properties, based on classification by Garrett & Grisham (2010) with some modifications (cp. Appendix Section A.3). In this modification of our method, which we refer to as BetterBodies-C(ontrastive), we added a supervised contrastive loss to the VAE training objective to realize this grouping in latent space (Khosla et al., 2020).

Specifically, the contrastive loss is given by

$$-\sum_{a \in A} log\Big[\frac{1}{|\text{group}(a)|} \sum_{p \in \text{group}(a)} \frac{\exp(z_a \cdot z_p)/\tau}{\sum_{a' \in A \setminus a} \exp(z_a \cdot z'_a)/\tau}\Big], \quad (3)$$

where $A$ is the set of all AAs, group($a$) represents the subset of AAs belonging to the same group as $a$, $\cdot$ represents the cosine similarity over latent representations $z$, and $\tau$ is a temperature hyperparameter. This loss maximizes the similarity between AAs in the same group and maximizes it in between groups.

Recall from Section 2, that the diffusion's reverse process starts with $x^N \sim \mathcal{N}(x^N; 0, \mathbf{I})$. Consequently, the regularization term KL($\mathcal{N}(\mu_\omega^a, \sigma_\omega^a), \mathcal{N}(0, \mathbf{I})$) also prevents a mismatch between the AA latent space and the diffusion model's latent space.

## 4.2 GUIDING DIFFUSION POLICIES USING REINFORCEMENT LEARNING

The policy $\pi$ is trained to achieve a balance between two objectives: generating latent vectors representing AAs with high likelihood given a dataset $D$ and generating AAs maximizing a learned Q-function. Recall from Section 2.4, that we use $a$ to represent an AA, its corresponding latent vector, and the corresponding action in the MDP.

The loss function corresponding to the first objective, referred to as the behavior cloning (BC) loss, is a slight adaptation of the standard loss function for continuous diffusion models given in Equation 1. In particular, as we generate sequences stepwise, one AA $a$ after the other, we condition the diffusion model on the sequence $s$ of AAs generated so far. The first loss function thus becomes

$$L_{BC}(\theta) = \mathbb{E}_{n \sim \text{Unif}(1,N), \epsilon \sim \mathcal{N}(0,\mathbf{I}), (s,a) \sim D}[||\epsilon - \epsilon_\theta(\sqrt{\bar{\alpha}^n}a + \sqrt{1 - \bar{\alpha}^n}\epsilon, s, n)||^2]. \quad (4)$$

Simply put, this loss function trains the model to reconstruct the next $a$ given an incomplete sequence $s$ from the dataset $D$. It has been shown that this diffusion approach improves performance on multimodal data in comparison to other training paradigms (Wang et al., 2023).

The BC loss alone does not provide a means of generating AAs which would result in sequences with improved affinity compared to sequences in $D$. Consequently, we desire a gradient guiding the policy $\pi_\theta$ towards such AAs. We follow Wang et al. (2023) and utilize the gradient of a learned Q-function $Q(s, a^0)$ given an incomplete sequence $s$ and an action $a^0$ generated by the policy $\pi_\theta$. The use of a Q-function for guidance in sequence design is promising, as these have been shown to stitch together improved sequences from suboptimal ones and excel in states where taking a specific action is required (Kumar et al., 2022). The full loss for $\pi_\theta$, represented by its learnable parameters $\theta$ is then given as

$$L(\theta) = L_{BC}(\theta) - \eta \cdot \mathbb{E}_{s \sim D, a^0 \sim \pi_\theta}[Q_\phi(s, a^0)]. \quad (5)$$

As $a^0$ is generated using the reverse process of the diffusion model $\pi_\theta$, the gradient of $Q_\phi(s, a^0)$ is propagated through the diffusion model's reverse process, thereby guiding the selection of actions with a high Q-value given the current state $s$. The hyperparameter $\eta$ is used to select a balance between the BC loss and maximizing Q-values. This relatively straightforward combination of a loss function that regularizes the policy towards the dataset and a loss function that facilitates policy improvement beyond the dataset has been demonstrated to be effective in many offline RL domains (Fujimoto & Gu, 2021).

The Q-function $Q_\phi$, implemented as clipped double Q-learning (Fujimoto et al., 2018), is trained to minimize the so-called TD-error:

$$\mathbb{E}_{(s_t, a_t, s_{t+1}) \sim D, a_{t+1}^0 \sim \pi'_\theta}[||(R(s_t, a_t) + \min_{i=1,2} Q_{\phi'_i}(s_{t+1}, a_{t+1}^0)) - Q_{\phi_i}(s_t, a_t)||^2], \quad (6)$$

where subscripts $t$ indicate the trajectory index (AA position). In practice, the diffusion policy $\pi_\theta$ and the Q-function $Q_\phi$ are updated in an alternating fashion.

## 4.3 FILTERING GENERATED SEQUENCES

Finally, we propose a filtering method to enhance the average affinity of returned sequences, referred to as BetterBodies-F(iltering). Consequently, we refer to our method with both filtering and contrastive latent as BetterBodies-CF. As stated in Section 2.4, we only assign a reward corresponding

to the sequence's free energy to full sequences of length 11. Consequently, the Q-value $Q_\phi(s_{10}, a_{10})$ of a sequence of length 10 $s_{10}$ concatenated with the last amino acid $a_{10}$ is trained to predict the free energy. We propose to utilize the learned Q-values as a discriminator and sort generated sequences according to their predicted free energy. This allows for the discarding of high-energy sequences above a given percentile. If the learned Q-values do correlate with the true affinity (inverse of free energy), this method will be effective in retaining high-affinity sequences.

## 5  EXPERIMENT SETUP

In the following section, we compare our method to GFlowNets (Jain et al., 2022), using the hyper-parameters provided by the authors, on three different data distributions. Further, results for a second antigen and implementation details are included in the appendix, Section A.1 and Section A.2. Note, that the utilized datasets and our source code will be included in the supplementary material upon publication.

**Evaluation Metrics**  Our objective is to train a policy $\pi$ which generates a set of unique novel sequences, denoted $D_{gen}$, given a training dataset $D$. The novel sequences should maximize binding affinity to a given antigen. Affinity can be maximized by minimizing the free energy between the antibody, represented by the generated sequence, and the antigen. Therefore, we want to minimize the free energy computed using the Absolut! software. Furthermore, generated sequences should be diverse and novel in comparison to the dataset $D$. We utilize the definition of diversity and novelty proposed by Jain et al. (2022): $Diversity(D_{gen}) := \frac{\sum_{x_i \in D_{gen}} \sum_{x_j \in D_{gen} \setminus \{x_i\}} d(x_i, x_j)}{|D_{gen}|(|D_{gen}|-1)}$ and $Novelty(D_{gen}) := \frac{\sum_{x_i \in D_{gen}} \min_{s_j \in D} d(x_i, s_j)}{|D_{gen}|}$, where $d(\cdot, \cdot)$ is the Levenshtein distance measuring the amount of difference between two sequences (Miller et al., 2009). These measures provide insight into the average number of pointwise mutations in the sequence relative to other sequences in the generated dataset $D_{gen}$ (Diversity) and their closest relative in the original dataset $D$ (Novelty).

**Training Datasets**  To assess the efficacy of our method we train it on three different data distributions. These distributions represent CDRH3 sequences and their respective free energies in complex with the SARS-CoV spike receptor-binding domain (PDB ID 2DD8_S). We selected this antigen, as prior methods on the Absolut! benchmark (Cowen-Rivers et al., 2022; Khan et al., 2022; Vogt et al., 2023) performed comparably weak on this antigen, indicating a higher complexity in identifying effective binders. We present additional evaluations on a second target in the supplementary material. The first distribution comprises a set of 2500 randomly generated sequences, for which the binding affinity to the SARS-CoV spike receptor-binding domain was predicted using the Absolut! software. The second set contains 2753 murine CDRH3 sequences, which were categorized as good but not exceptional binders Robert et al. (2022). The final distribution, comprising 2167 sequences, was gathered during the exploration phase of a Q-learning agent, similar to those described by Vogt et al. (2023). Due to the agent's efficacy, this dataset contains sequences that reach affinity levels beyond those found with murine CDRH3 sequences. We refer to the three datasets as *random*, *natural*, and *expert*. The three datasets thereby represent data distributions that could occur in applications of our method. The random data represents an initial lab screening with random CDRH3 sequences, natural a dataset derived from known natural sequences, and expert a dataset as it could occur in an active-learning setting.

## 6  RESULTS

In the following section, we present the results of our experiments. To develop an intuition, we start with an in-depth analysis of the effect of Q-function guidance on diffusion with respect to the maximization of the ELBO given the dataset $D$, the training stability, as well as the affinity, novelty, and diversity of generated sequences on the expert dataset. Subsequently, we compare our method to GFlowNets (Jain et al., 2022) on all three datasets. All experiments are carried out over five seeds.

### 6.1 Effect of Guidance

Our method augments the generative process of a diffusion model with Q-value guidance. To achieve an improvement in performance over basic diffusion, it is necessary to find an appropriate balance between two objectives: optimizing the ELBO (represented by $L_{BC}$) given data $D$ and maximizing the Q-value. As discussed in Section 4, Equation 5, this balance can be controlled using hyperparameter $\eta$, where $\eta = 0$ deactivates the guidance leading to basic diffusion only optimizing the ELBO.

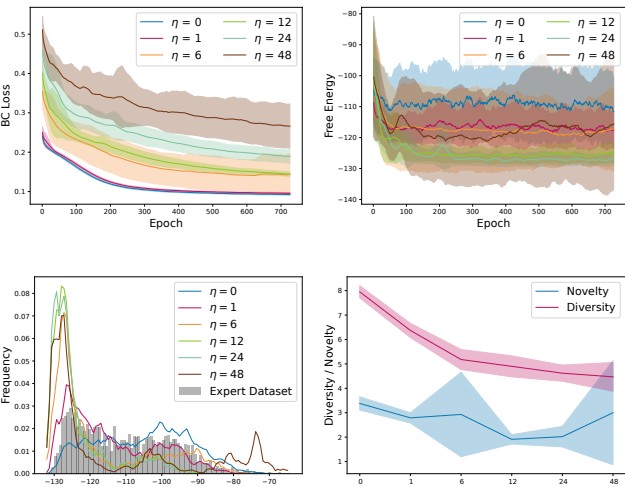

Figure 2: The effect of various $\eta$ settings: On the basic diffusion loss $L_{BC}$ (top left), free energy evaluated during training (top right), and free energy distribution of generated unique novel sequences (bottom left), and Diversity and Novelty of generated sequences (bottom right). Distributions of generated sequences are plotted as a running average over three bins.

In Figure 2 (top left), we visualize the effect of varying $\eta$ configurations on the magnitude of $L_{BC}$. We observe an increasing trend in $L_{BC}$ when increasing $\eta$, suggesting a shift of the policy away from $D$.

This shift, up to a certain point, corresponds to an increase in the affinity of generated sequences during the training phase, as illustrated in Figure 2 (top right). However, with $\eta = 48$ training instabilities can be observed.

Figure 2 (bottom left), depicts the free energy distribution of unique novel sequences generated after the training phase across multiple $\eta$ settings. While $\eta = 0$ roughly matches the training distribution, increasing $\eta$ up to 24 results in a shift of the distribution towards sequences with low free energy, highlighting the improvement through guidance.

In Figure 2 (bottom right) we can visualize the influence of $\eta$ on the diversity and novelty of generated sequences. With increasing choice of $\eta$, diversity is decreasing, indicating a guidance towards a narrow distribution of sequences, maximizing the Q-function. For novelty, we can observe a similar trend, which, however, stops with the unstable setting of $\eta = 48$ where novelty increases again.

### 6.2 Comparison to GFlowNets

Having demonstrated the efficacy of our method and the impact of balancing basic diffusion and Q-guidance, we now present results regarding multiple diverse data distributions. We selected $\eta = 24$ for our method, analyzing the effect of our filtering and contrastive latent method. We further compare our method to Basic Diffusion, where $\eta = 0$, and GFlowNets (Jain et al., 2022).

For the filtering method, we include sequences above the 50th affinity percentile, scored by the Q-function. Analogously, we apply a filtering step to the sequences generated by GFlowNets, including

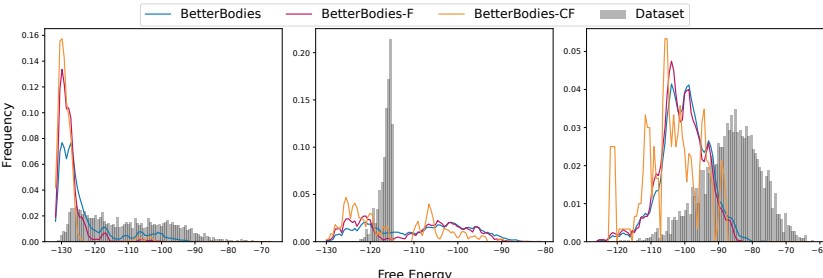

Figure 3: Free energy distributions of unique training dataset sequences and generated sequences. The random (left), natural (middle), and expert (right) datasets are visualized histograms. Sequences generated using BetterBodies $\eta = 24$, it's F(iltering), and C(ontrastive) versions are plotted as a running average over three bins. Data is visualized as the mean over five seeds.

the sequences above the 50th percentile scored by the method's own learned reward model. We generate 500 novel sequences per dataset, thus returning 250 sequences after filtering.

In Figure 3 we visualize the free energy distributions of sequences returned by our methods in comparison to the given dataset distribution. In Table 1 we give numerical results, comparing also to GFlowNets and giving an insight into the novelty and diversity of generated sequences. We can observe from the distributions that filtering and contrastive latent further shift the distributions of free energies towards low free energies, indicating an improved performance. This is further supported by the Free Energy scores provided in the tabular results. In Figure 4 we visualize how the contrastive latent changes the latent representations of AAs and the corresponding average Q-values. We can observe that the contrastive latent method allows to cluster AAs which on average lead to better affinity scores.

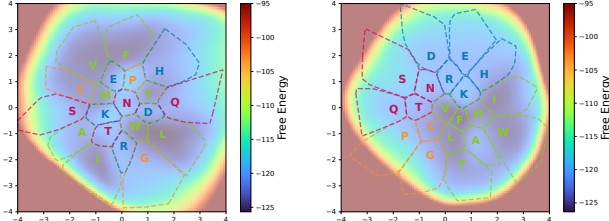

Figure 4: VAE latent space encoding amino acids, utilizing no regularization (left) and with contrastive loss regularization (right). Amino acid groups are indicated by the coloring and the space occupied by their samples. The underlying heatmap displays the average Q-value over 1000 sequence-action pairs.

We observe that our methods outperform or match the performance of GFlowNets regarding Free Energy on all three datasets. We can further observe that novelty and diversity tend to decrease alongside decreasing free energy. Nonetheless, sequence sets generated by GFlowNets exhibit a higher diversity and novelty when matching the free energy level of our methods. Additionally, we observe that both our methods and GFlowNets struggle to generate sequence sets whose mean free energy reaches that of the natural dataset, due to a large fraction of low-affinity binders (cp. Figure 3(middle)). We hypothesize that this is due to the narrow distribution and lack of low-affinity samples in the dataset. Interestingly, GFlowNets which samples actions proportional to their reward in the dataset struggles with the expert dataset, while Q-learning performs well, indicating an advantage in such settings. This coincides with findings by Shen et al. (2023), showing that low rewards were oversampled and GFlowNets failed to increase the expected reward despite scaled training rewards.

| | Method | Expert | Natural | Random |
|---|---|---|---|---|
| Free Energy | Dataset | -110.53 ± 12.84 | -116.46 ± 1.49 | -86.21 ± 8.75 |
| | Basic Diffusion | -105.02 ± 2.19 | -109.28 ± 1.29 | -84.74 ± 0.83 |
| | BetterBodies | -123.23 ± 2.45 | -108.40 ± 2.44 | -99.64 ± 2.64 |
| | BetterBodies-F | -127.44 ± 1.68 | -110.53 ± 2.50 | -100.55 ± 2.89 |
| | BetterBodies-CF | **-128.20 ± 0.30** | **-113.40 ± 1.57** | **-103.56 ± 3.29** |
| | GFlowNets | -103.85 ± 0.55 | -108.11 ± 0.37 | -101.28 ± 0.47 |
| | GFlowNets-F | -101.71 ± 0.68 | -108.98 ± 0.29 | **-104.02 ± 0.43** |
| Diversity | Dataset | 7.72 | 7.38 | 10.27 |
| | Basic Diffusion | 7.95 ± 0.25 | 8.00 ± 0.22 | 10.17 ± 0.01 |
| | BetterBodies | 4.62 ± 0.33 | 6.16 ± 0.70 | 5.06 ± 0.81 |
| | BetterBodies-F | 4.23 ± 0.35 | 5.60 ± 0.73 | 4.55 ± 0.79 |
| | BetterBodies-CF | 4.22 ± 0.22 | 5.36 ± 0.70 | 6.38 ± 0.75 |
| | GFlowNets | 9.20 ± 0.13 | 6.08 ± 0.09 | 9.24 ± 0.08 |
| | GFlowNets-F | 9.14 ± 0.07 | 5.60 ± 0.13 | 8.77 ± 0.09 |
| Novelty | Basic Diffusion | 3.38 ± 0.27 | 2.68 ± 0.18 | 6.37 ± 0.04 |
| | BetterBodies | 2.02 ± 0.42 | 2.89 ± 0.44 | 6.24 ± 0.86 |
| | BetterBodies-F | 1.82 ± 0.40 | 2.66 ± 0.40 | 6.10 ± 1.15 |
| | BetterBodies-CF | 1.50 ± 0.07 | 2.59 ± 0.45 | 6.01 ± 0.78 |
| | GFlowNets | 6.30 ± 0.07 | 5.99 ± 0.06 | 6.62 ± 0.02 |
| | GFlowNets-F | 6.29 ± 0.05 | 5.95 ± 0.06 | 6.63 ± 0.04 |

Table 1: Free energy, diversity, and novelty of sequences generated by our method, $\eta = 24$, the filtering and contrastive latent method in comparison with Basic Diffusion and GFlowNets, on the expert, natural, and random dataset. Best performing free energy values are written in bold.

## 7 CONCLUSION

We presented BetterBodies a novel method for antibody CDRH3 sequence design, demonstrating the applicability of guided latent diffusion for successive AA sequence design. Our method successfully generates novel, diverse, and high-affinity sequences towards the SARS-CoV spike receptor-binding domain given three different sequence and affinity distributions, evaluated using the Absolut! software. We demonstrated that Q-value guidance and our novel filtering and contrastive latent methods enhance the affinity of generated sequences when compared to basic diffusion. We further demonstrate that our methods match or exceed the affinity scores of GFlowNets, but sometimes generates less diverse sequence sets. In conclusion, methods such as ours have the potential to have great implications for real-world biological sequence design, where the generation of novel high-affinity binders is a cost-intensive endeavor (Norman et al., 2020; Shanehsazzadeh et al., 2023).

## 8 LIMITATIONS AND FUTURE WORK

In this work, we proposed a novel method for protein sequence generation using diffusion models and RL. One of the main drawbacks of diffusion models is the relatively high computational time, especially for higher $N$. This could presumably be reduced using methods by Kang et al. (2024), Nichol & Dhariwal (2021), or Song et al. (2021), which would increase the training and inference speed of our method. Additionally, there are many recent methods proposed in the offline RL community (Levine et al., 2020) which could be used instead of clipped double Q-learning (Fujimoto et al., 2018). Finally, our method could be extended to the model-based and active learning setting and subsequently be evaluated using the sequence tasks proposed by Jain et al. (2022) and Trabucco et al. (2022).

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

# A APPENDIX

## A.1 RESULTS ON A SECOND ANTIGEN

We carried out additional experiments designing antibody CDRH3 sequences binding the human CD38 (PDB ID: 3RAJ_A), also known as cyclic ADP ribose hydrolase. For simplicity, we reference the antigen by its PDB ID. The datasets were retrieved as for the experiments on 2DD8, leading to datasets of size 2500, 5463, and 2103 respectively. In Table 2 we present the corresponding results.

| | Method | Expert | Natural | Random |
|---|---|---|---|---|
| **Free Energy** | Dataset | -98.39 ± 12.13 | -106.09 ± 1.51 | -81.90 ± 8.06 |
| | Basic Diffusion | -93.66 ± 1.61 | -97.77 ± 0.81 | -80.69 ± 0.98 |
| | BetterBodies | -107.98 ± 5.61 | -103.59 ± 2.91 | -92.97 ± 2.94 |
| | BetterBodies-F | **-113.48 ± 6.35** | -107.13 ± 3.31 | -94.04 ± 3.58 |
| | BetterBodies-CF | -110.33 ± 8.87 | **-110.39 ± 1.03** | -94.36 ± 2.96 |
| | GFlowNets | -93.46 ± 2.62 | -101.21 ± 0.66 | -94.58 ± 0.37 |
| | GFlowNets Filtered | -94.79 ± 3.21 | -104.77 ± 0.89 | **-96.27 ± 0.22** |
| **Diversity** | Dataset | 8.30 | 8.06 | 10.27 |
| | Basic Diffusion | 8.27 ± 0.12 | 8.23 ± 0.21 | 10.17 ± 0.01 |
| | BetterBodies | 4.91 ± 0.78 | 5.37 ± 0.36 | 6.07 ± 0.55 |
| | BetterBodies-F | 4.53 ± 0.66 | 4.64 ± 0.46 | 5.48 ± 0.66 |
| | BetterBodies-CF | 4.21 ± 0.33 | 4.36 ± 0.40 | 5.46 ± 0.70 |
| | GFlowNets | 8.33 ± 0.19 | 4.69 ± 0.18 | 9.29 ± 0.07 |
| | GFlowNets-F | 8.08 ± 0.22 | 4.28 ± 0.24 | 8.90 ± 0.09 |
| **Novelty** | Basic Diffusion | 3.61 ± 0.14 | 2.80 ± 0.22 | 6.37 ± 0.04 |
| | BetterBodies | 2.38 ± 0.63 | 2.82 ± 0.29 | 6.32 ± 0.41 |
| | BetterBodies-F | 2.18 ± 0.68 | 2.50 ± 0.24 | 6.18 ± 0.61 |
| | BetterBodies-CF | 2.66 ± 1.71 | 2.43 ± 0.53 | 5.07 ± 1.12 |
| | GFlowNets | 5.76 ± 0.10 | 4.55 ± 0.10 | 6.63 ± 0.01 |
| | GFlowNets-F | 5.64 ± 0.10 | 4.50 ± 0.10 | 6.63 ± 0.03 |

Table 2: Antigen 3RAJ_A; Free energy, diversity, and novelty of sequences generated by our method, $\eta = 24$, the filtering and contrastive latent method in comparison with Basic Diffusion and GFlowNets, on the expert, natural, and random dataset. Best performing free energy values are written in bold.

## A.2 IMPLEMENTATION DETAILS

To reduce the effect of the latent space' structure on the reported results, we share the pre-trained VAE between all datasets for a given seed. Due to the large computational burden, we chose $N = 5$ diffusion steps for our experiments, even though we found that $N = 50$ leads to better results for $\eta = 0$. This finding is analogous to (Wang et al., 2023). We follow (Wang et al., 2023) for the choice of $\beta$ noise schedule to train our diffusion model.

As in the implementation by Wang et al. (2023) we generate 50 actions using the Diffusion Model per step and sample the final action via a softmax distribution over the respective Q-weights.

Note, that we choose to represent $s$ for the Policy $\pi$ and Q-function as a concatenation of one-hot encodings, which represent the previous AAs, due to its simplicity. In theory, a concatenation of VAE latent vectors, or a latent vector representing the entire sequence, could also be used.

## A.3 AMINO ACID GROUPS

Our grouping of AAs is mostly based on work by Garrett & Grisham (2010) with the following modifications:

- We add the "Special Case" group

- we classify "P" as a special case as it "is not an amino acid but rather an $\alpha$-imino acid."Garrett & Grisham (2010) and 'its unusual cyclic structure"Garrett & Grisham (2010).

- we classify "G" as a special case as it does not have a side chain.

- we classify "C" as a special case as it can "deprotonate at pH values greater than 7"Garrett & Grisham (2010).

- we classify "Y" as hydrophobic as Garrett & Grisham (2010) argue that it could also be classified as such.

Note, that we chose this specific grouping not because we are convinced it bears an advantage, but rather because it was the first grouping we found in literature.

