# OpenReview forum: "BetterBodies: Reinforcement Learning guided Diffusion for Antibody Sequence Design"
_ICLR.cc/2025/Conference — ICLR 2025 Conference Withdrawn Submission_

### Official Review · Reviewer_H4CR · 2024-10-28

**Soundness:** 1
**Presentation:** 2
**Contribution:** 1
**Rating:** 3
**Confidence:** 5

**Summary:**

This paper proposes BetterBodies, a framework that trains diffusion policy with RL to generate high-affinity CDRH3 sequences. They use the VAE to continuously represent the sequences and evaluate their results with Absolut! simulator.

**Strengths:**

The method itself is easy to understand and the use of diffusion models and RL is well-motivated.

**Weaknesses:**

This paper brings up several points worth considering:
- **Motivation**: The decision to use a 2-dimensional latent space in the VAE could be more clearly explained. While the authors suggest that combining VAE with contrastive learning can help group amino acids, it would be helpful to understand more specifically how this approach impacts model performance. Additionally, the rationale for a 2-dimensional space might benefit from further justification, as it may limit the ability to capture complex relationships. It seems that balancing high likelihood with predicted affinity might be possible using discrete diffusion alone, so more clarification on the essential role of the VAE component would be valuable.
- **Novelty**: The proposed method draws on a combination of existing techniques. A more detailed discussion on how these elements uniquely contribute to antibody design could enhance the sense of novelty.
- **Related Work**: Expanding the related work section to include notable antibody research from recent ML conferences (such as dWJS [1], AbODE [2], etc.) would strengthen the contextual foundation and illustrate the paper’s position within the field.
- **Experiment**: The experimental setup might benefit from including additional, recent baselines beyond GFlowNet [3], as several antibody-specific generative models have emerged in the past two years. Also, considering other evaluation tools like Rosetta or ML-based methods common in antibody research could provide more robust benchmarking. The Absolut! framework limits the evaluation to sequences of 11 residues, which might narrow the scope unnecessarily. Given these considerations, the current comparison could be enriched by including recent models or broader evaluation metrics.
- **Writing**: The clarity could be improved by refining certain sections for conciseness. For example, the discussion on diffusion models in the opening of Section 4 could be streamlined to maintain focus on the methodology itself.

---
[1] Protein Discovery with Discrete Walk-Jump Sampling, Frey et al., ICLR 24
[2] AbODE: Ab Initio Antibody Design using Conjoined ODEs, Verma et al., ICML 23
[3] Biological sequence design with gflownets, Jain et al., ICML 22

**Questions:**

1. Could the authors clarify the necessity of the VAE component and the reasoning behind selecting a 2-dimensional latent space?
2. Given the mentioned limitations, what were the considerations for choosing Absolut! software for evaluation?
3. Is there a reason prominent antibody papers from recent ML conferences weren't cited or compared? Many of these also provide accessible open-source implementations.

---

> ### Author Response · Authors · 2024-11-19
>
> We thank Reviewer H4CR for their constructive feedback.
>
>
>
> As pointed out by the reviewer, the use of diffusion models and RL is well-motivated, however it appears that in the current version of the manuscript we did not well motivate the use of VAE. We thus agree with the reviewer's opinion that an ablation on the use of VAE's could be added, e.g. by comparing to a discrete diffusion model, and the reason for usage of a VAE could be more thoroughly explained.
>
> Like Reviewer LSPg, Reviewer H4CR suggest adding more baselines and evaluation metrics. While we stand by our choice of selecting GFlowNets as the main baseline, due to it being frequently cited and many GFlowNet-based methods being published, adding more baselines could improve the manuscript by facilitating comparison. Regarding evaluation metrics, we think that the limitations chosen by Absolut! are reasonable, as the CDRH3 is the most relevant for binding properties [1,2] and a sequence length of 11 is realistic for CDRH3 sequences [3,4]. Nonetheless, adding more evaluation metrics could improve the manuscript. We explicitly did not choose an ML-based evaluation metric to prevent our method from exploiting erroneous local minima, which can occur during the ML-extrapolation process.
>
>
>
> [1] R. A. Norman et al., “Computational approaches to therapeutic antibody design: established methods and emerging trends,” Brief Bioinform, vol. 21, no. 5, pp. 1549–1567, Sep. 2020
>
> [2] J. L. Xu and M. M. Davis, “Diversity in the CDR3 Region of VH Is Sufficient for Most Antibody Specificities,” Immunity, vol. 13, no. 1, pp. 37–45, 2000,
>
> [3] C. Joyce, D. R. Burton, and B. Briney, “Comparisons of the antibody repertoires of a humanized rodent and humans by high throughput sequencing,” Sci Rep, vol. 10, no. 1, p. 1120, Jan. 2020
>
> [4] X. Chen, M. Gentili, N. Hacohen, and A. Regev, “A cell-free nanobody engineering platform rapidly generates SARS-CoV-2 neutralizing nanobodies,” Nat Commun, vol. 12, no. 1, Art. no. 1, Sep. 2021

---

> > ### Comment · Reviewer_H4CR · 2024-11-23
> >
> > Thank you for your response. My key concerns remain unresolved:
> > - The motivation for using VAE is still unclear.
> > - The frequent citation of GFlowNet is not a sufficient justification for excluding other baselines.
> > - While the author agrees that adding more evaluation metrics could improve the work, no additional experiments were conducted.
> > - My concern regarding the related work section has not been addressed.
> >
> > For these reasons, I will maintain my current rating.

---

### Official Review · Reviewer_b3dp · 2024-11-03

**Soundness:** 2
**Presentation:** 2
**Contribution:** 2
**Rating:** 5
**Confidence:** 4

**Summary:**

In this paper, the authors introduce BetterBodies, a method using variational autoencoders and reinforcement learning to generate antibody CDR H3 libraries. They use a VAE to compress amino acids to a two-dimensional latent space, grouped by physicochemical properties using a contrastive loss.

**Strengths:**

The paper engages with a complex and highly impactful problem setting: designing therapeutic antibodies.

**Weaknesses:**

The paper demonstrates a lack of awareness of both the practicalities of the problem at hand (ML-enabled antibody design) and the state-of-the-art in the field. Statements like “This [RL] is well suited to antibody sequence design, where direct access to a wet lab is not feasible” and “matches or outperforms the state-of-the-art method Generative Flow Network (GFlowNet)” reveal basic misunderstandings about the problem setting and state of the field.
Developing methods for the offline setting using RL is not particularly relevant to real-world protein design, but is still of methodological interest and is interesting to explore, given contextualization with the broader field and the severe and impractical limitations of operating in an offline setting. There is already a rich literature in the online setting for protein and antibody design, which should be referenced and contrasted against. One such example is Li, Lin, et al. "Machine learning optimization of candidate antibody yields highly diverse sub-nanomolar affinity antibody libraries." Nature Communications 14.1 (2023): 3454. The main comparison is to GFlowNets, which is relevant insofar as the paper considers the offline RL setting, but the authors do not discuss or benchmark against any methods that are actually used in antibody design (i.e., methods that include wet lab validation).
Antibodies are a leading class of drugs, but not only (or mainly) because of their binding properties.

**Questions:**

Can the authors provide sets of generated sequences and sequence-level evaluation beyond diversity and novelty to show their method is not simply reward hacking the Absolut! benchmark?

---

> ### Author Response · Authors · 2024-11-19
>
> We thank Reviewer b3dp for their feedback to which we response below.
>
> The reviewer suggests a "lack of awareness of both the practicalities of the problem at hand (ML-enabled antibody design) and the state-of-the-art in the field" highlighting that "Developing methods for the offline setting using RL is not particularly relevant to real-world protein design". We assume that there is a discrepancy between our definition of "offline RL" and the reviewers. We use the definition as for example in [1], where offline defines "algorithms that utilize previously collected data, without additional online data collection" in contrast to online where the RL agent repeatedly, often for thousands of iterations, interacts with the environment during training. Further, there are model-based approaches, compatible with both the online or offline paradigm, where a learned dynamics/reward function is used instead or in addition to online interaction or learning purely from pre-collected datasets.
>
> We stand by our assessment, backed by many previous works [2, 3, 4], that a repeated interaction with a wet lab is not feasible at the current state-of-the-art. We assume the reviewer agrees to this point.
>
> To facilitate a scientific discussion, we would therefore ask the reviewer to provide their definition of an offline vs an online algorithm.
>
> Furthermore, we investigated reference [5] kindly provided by the reviewer. The ML-Method proposed therein can be summarized as follows: a) a dataset is created using real-world experiments b) a "sequence-to-affinity model" is trained offline on the dataset and publicly available data c) Bayesian optimization over the learned model is used to generate a novel dataset d) the novel dataset is evaluated in silico e) the top binders according to in silico validation are experimentally validated.
>
> According to the definition of [1] the method proposed in [5] is therefore a model-based offline algorithm, due to having no online interaction with the environment during training. The core difference with respect to the setting chosen in our manuscript is that the authors of [5] had access to a wet lab for step a) and e) where we instead used the Absolut! framework. However, this has no influence on the working mechanisms of the ML approach b), c) and d). Indeed, the workflow of our method is rather similar, where our method learns directly from the data, removing step b), and using our RL-based method instead of Bayesian optimization in step c). Instead of the in silico evaluation in step d) we used our proposed filtering method.
>
> The reviewer further asks to provide sets of generated sequences to show that the "method is not simply reward hacking the Absolut! benchmark". Those sequences would of course be included during code publication. However, the authors are not sure what exactly the reviewer means by "reward hacking" in this case and ask for clarification.
>
>
>
> [1] S. Levine, A. Kumar, G. Tucker, and J. Fu, “Offline Reinforcement Learning: Tutorial, Review, and Perspectives on Open Problems,” Nov. 01, 2020
>
> [2] C. Angermueller, D. Dohan, D. Belanger, R. Deshpande, K. Murphy, and L. Colwell, “Model-based reinforcement learning for biological sequence design,” presented at the International Conference on Learning Representations, May 2020.
>
> [3] M. Jain et al., “Biological Sequence Design with GFlowNets”.
>
> [4] N. Gruver et al., “Protein Design with Guided Discrete Diffusion,” May 31, 2023
>
> [5] L. Li et al., “Machine learning optimization of candidate antibody yields highly diverse sub-nanomolar affinity antibody libraries,” Nat Commun, vol. 14, no. 1, p. 3454, Jun. 2023

---

> > ### Comment · Reviewer_b3dp · 2024-11-24
> >
> > We are in agreement on the definition of offline vs online. I do not agree with the statement "a repeated interaction with a wet lab is not feasible at the current state-of-the-art". In fact, the state-of-the-art is completely defined by online methods. The authors themselves cite papers that include repeated interactions with the wet lab. Using simulations or synthetic test functions is a perfectly valid way to operate in the online setting while under resource constraints.
> >
> > I am in full agreement with Reviewer H4CR that "The frequent citation of GFlowNet is not a sufficient justification for excluding other baselines." The authors state that "Yoshua Bengio alone published more than 30 papers regarding GFlowNets...", which is not a scientific argument. I would encourage the authors to read more broadly in the field, and look for work that engages specifically with scientific problems, rather than relying on appeals to authority. After considering the author responses and other reviews, I will maintain my score.

---

### Official Review · Reviewer_LSPg · 2024-11-03

**Soundness:** 2
**Presentation:** 2
**Contribution:** 1
**Rating:** 3
**Confidence:** 4

**Summary:**

In this paper, the authors propose a framework to generate CDRH3 sequences with improved binding affinities to important receptors. In particular, the framework first pre-trains a variational autoencoder (VAE) to map each amino acid in CDRH3 sequences to a latent embedding. It then learns a diffusion model that sequentially generates these embeddings, which can be transformed back to amino acids, to construct the full CDRH3 sequences. To further improve the binding affinity, a Q function is trained to accurately estimate the Q-value for each action (i.e., each added amino acid), with previously added amino acids. This Q function is also used to guide the training of the diffusion model for better binding affinities. The experimental results show that the model can match or outperform the baseline.

**Strengths:**

* The method combines a VAE, a diffusion model, and reinforcement learning together to generate CDRH3 sequences.
* The method can match or outperform the baseline.

**Weaknesses:**

* The formulated problem is relatively simple and may not require such a complex framework. In this paper, only CDRH3 sequences with 11 amino acids are considered, and variations in sequence length are not considered. With each amino acid having only 20 amino types, the search space, in the reviewer’s opinion, is limited. Given that specific motifs may exist at different positions in CDRH3, a combination of three complicated models may not be necessary for this task.

* It is suggested to include other state-of-the-art baselines for biological sequence design in the comparison. Currently, the paper uses only one baseline, which is not sufficient.

* It is suggested to clearly explain the design choices and do extensive ablation studies to validate them. For example, it would be helpful to validate the importance of the VAE latent space and the use of diffusion models in the overall framework.

**Questions:**

* The reviewer thinks this task is relatively simple. Given this simplicity, the reviewer thinks this complicated framework may not be necessary.  To validate the effectiveness of this framework, it is suggested that the authors add some simple search or optimization algorithms as baselines. For example, genetic algorithms, MCTS search algorithms, or even random policies, can also be used to generate novel CDRH3 sequences or mutate existing CDRH3 sequences for improved binding affinities. The reviewer wonders, within the same time constraints, whether these search or optimization algorithms can perform comparable or better than the proposed framework.

* The rationale for using a VAE to learn a two-dimensional latent vector for each amino acid is unclear. Given that the input sequence has a fixed length of 11, the reviewer thinks it is not necessary to learn additional embeddings for each amino acid.

* The rationale for using a diffusion model to generate each amino acid also requires further explanation. The reviewer thinks that determining the amino acid type based on previously generated sequences is not a complicated problem, especially given the short length of sequences and limited number of amino acid types.


* It is suggested that more baselines [1,2] need to be added for a comprehensive comparison.  Some related works [3, 4] for CDR3 sequences should also be discussed.

[1] Angermueller, Christof, et al. "Model-based reinforcement learning for biological sequence design." International conference on learning representations. 2019.

[2] Chen, Can, et al. "Bidirectional learning for offline model-based biological sequence design." International Conference on Machine Learning. PMLR, 2023.

[3] Chen, Ziqi, et al. "T-cell receptor optimization with reinforcement learning and mutation polices for precision immunotherapy." International Conference on Research in Computational Molecular Biology. Cham: Springer Nature Switzerland, 2023.'

[4] Karthikeyan, Dhuvarakesh, et al. "Conditional Generation of Antigen Specific T-cell Receptor Sequences." NeurIPS 2023 Generative AI and Biology (GenBio) Workshop.

---

> ### Author Response · Authors · 2024-11-19
>
> We thank Reviewer LSPg for their feedback to which we response below.
>
> The reviewer mentions that "this task is relatively simple" and "the search space, in the reviewer’s opinion, is limited". Given a sequence length of 11 amino acids and 20 possible amino acids per position, the considered search space comprises ~205 trillion sequences (see section 2.1). Of course, longer sequences will result in even larger search spaces, but we respectfully disagree with the reviewer's opinion that 205 trillion sequences are a limited amount. Furthermore, we deliberately chose to only design the CDRH3 region due to it being the most relevant for binding [1, 2].
>
> The CDRH3 length of 11, chosen in the Absolut! Benchmark, is thereby within the range of frequently occurring CDRH3 lengths in antibodies [3] and nanobodies [4], representing a realistic setting for CDRH3 design. Designing full antibody sequences, including the design of framework regions which are highly conserved in nature [2,4], appears ineffective and is also typically not done in previous works [5] and real-world experiments [4].
>
> Therefore, while it would be interesting to evaluate the performance of our method on longer sequences, it is not directly relevant for the chosen task of antibody design.
>
> We agree with the reviewer's opinion on adding more baselines. Simple baselines would highlight necessity of advanced methods.
>
> With respect to state-of-the-art methods, we deliberately chose GFlowNets due to it being frequently cited (currently 155 citations) and a plethora of GFlowNet-based papers being published in recent years (Yoshua Bengio alone published more than 30 papers regarding GFlowNets and their application in various domains since 2022).
> While other recently published baselines would be interesting, we think that GFlowNets do well represent the state-of-the-art.
>
> The authors will try to improve their writing to facilitate understanding why design choices were made.
>
>  [1] R. A. Norman et al., “Computational approaches to therapeutic antibody design: established methods and emerging trends,” Brief Bioinform, vol. 21, no. 5, pp. 1549–1567, Sep. 2020
>
> [2] J. L. Xu and M. M. Davis, “Diversity in the CDR3 Region of VH Is Sufficient for Most Antibody Specificities,” Immunity, vol. 13, no. 1, pp. 37–45, 2000
>
> [3] C. Joyce, D. R. Burton, and B. Briney, “Comparisons of the antibody repertoires of a humanized rodent and humans by high throughput sequencing,” Sci Rep, vol. 10, no. 1, p. 1120, Jan. 2020
>
> [4] X. Chen, M. Gentili, N. Hacohen, and A. Regev, “A cell-free nanobody engineering platform rapidly generates SARS-CoV-2 neutralizing nanobodies,” Nat Commun, vol. 12, no. 1, Art. no. 1, Sep. 2021
>
> [5] W. Jin, J. Wohlwend, R. Barzilay, and T. Jaakkola, “Iterative Refinement Graph Neural Network for Antibody Sequence-Structure Co-design,” Jan. 27, 2022

---

> > ### Comment · Reviewer_LSPg · 2024-11-25
> >
> > Thank you for your response. However,  like the other reviewers, I feel my concerns remain unresolved.
> >
> > 1. Using only GFlowNet as a baseline is not enough. Including additional baselines, as also suggested by other reviewers, is essential.
> > 2. The rationale for choosing VAE and diffusion models remains unclear.
> >
> > Therefore, I will remain my current rating.

---

### Note · Authors · 2024-11-26

**Comment:**

After considering all the reviews, we agree that our manuscript would benefit from further experiments and clarification and hence, we withdraw from ICLR. We thank all the reviewers for the valuable feedback and the effort spent in reviewing our paper.

**Withdrawal Confirmation:**

I have read and agree with the venue's withdrawal policy on behalf of myself and my co-authors.